# A Multi-Robot Task Allocation Method Based on the Synergy of the K-Means++ Algorithm and the Particle Swarm Algorithm

**DOI:** 10.3390/biomimetics9110694

**Published:** 2024-11-13

**Authors:** Youdong Yuan, Ping Yang, Hanbing Jiang, Tiange Shi

**Affiliations:** School of Electromechanical Engineering, Lanzhou University of Technology, Lanzhou 730050, China; yangping_lz@163.com (P.Y.); 222080204004@lut.edu.cn (H.J.); 222085501039@lut.edu.cn (T.S.)

**Keywords:** multi-robot, particle swarm algorithm, task allocation, K-means++ clustering

## Abstract

Addressing challenges in the traditional K-means algorithm, such as the challenge of selecting initial clustering center points and the lack of a maximum limit on the number of clusters, and where the set of tasks in the clusters is not reasonably sorted after the task assignment, which makes the cooperative operation of multiple robots inefficient, this paper puts forward a multi-robot task assignment method based on the synergy of the K-means++ algorithm and the particle swarm optimization (PSO) algorithm. According to the processing capability of the robots, the K-means++ algorithm that limits the maximum number of clusters is used to cluster the target points of the task. The clustering results are assigned to the multi-robot system using the PSO algorithm based on the distances between the robots and the centers of the clusters, which divides the multi-robot task assignment problem into a multiple traveling salesmen problem. Then, the PSO algorithm is used to optimize the ordering of the task sets in each cluster for the multiple traveling salesmen problem. An experimental verification platform is established by building a simulation and physical experiment platform utilizing the Robot Operating System (ROS). The findings indicate that the proposed algorithm outperforms both the clustering-based market auction algorithm and the non-clustering particle swarm algorithm, enhancing the efficiency of collaborative operations among multiple robots.

## 1. Introduction

Multi-robot collaborative systems find extensive applications across diverse domains, including agriculture, post-disaster rescue operations, factory logistics, and the exploration of unknown environments [1,2,3,4,5]. This field has garnered significant attention and is among the most researched areas in robotics [6]. A crucial aspect of studying multi-robot systems is task allocation. The objective of task allocation in a multi-robot system is to devise a strategy that optimally assigns multiple tasks to different robots within the system. This allocation aims to minimize the overall task execution time, reduce travel distances to tasks, and maximize the completion rate of assigned tasks [7].

At present, domestic and foreign scholars’ research on task assignment in multi-robot systems mainly uses integer programming methods [8,9], market mechanism-based methods (MMA) [10,11], heuristic intelligent algorithms [12,13], and clustering-based task assignment algorithms [14,15]. Zou [16] introduced an enhanced assignment algorithm, building upon the Hungarian algorithm, aimed at boosting task execution efficiency and cutting down task execution costs within a task scheduling framework. Nonetheless, this method becomes computationally burdensome as the task volume escalates. On the other hand, Ren [17] presents a hybrid auction task allocation approach that integrates the K-means clustering algorithm with refinements in the selection of initial clustering centers. This refinement helps to minimize the robots’ traversal paths and overall system consumption. Despite the improvement in initial center selection, the process of clustering around these centers still necessitates manual intervention. Improper selection of cluster centroids can lead to increased path costs for the robots. Furthermore, the market-based task allocation strategy hinges on a robustly connected robot network, resulting in a lower completion rate when communication is disrupted or the communication environment is weak. Another approach is meta-heuristic algorithms such as the Genetic Algorithm (GA) [18] and the PSO algorithm [19], which are generalized methods for finding suboptimal solutions. Meta-heuristic methods can quickly find a solution, but the quality of the solution may be poor and sometimes falls into a local optimum. Therefore, the fusion of multiple heuristic algorithms provides a new way of thinking for solving the task allocation problem. Song [20] proposed a multi-robot task allocation method based on near-field subset partitioning to solve the problem of inefficiency in the distribution of medical supplies. The algorithm first utilizes the ant colony algorithm [21] to order the task set to form a chain of tasks related to the near field. Then, an objective optimization function is designed based on the task completion time and the path cost of the robots, and a genetic algorithm is used to divide the subsets of this task chain. Then, the divided subset of tasks is assigned to each robot. Finally, the effectiveness of the algorithm is verified by simulating an application scenario in a hospital ward. K-means clustering is also one of the most used methods for the task allocation problem. Janati et al. [22] first used K-means to group the tasks and then allocate them. Subsequently, they used the Genetic Algorithm (GA) to optimize the clustering results. Through simulation experiments, it can be observed that this method can effectively handle a large number of tasks and address the task allocation problem for robots. However, this method does not consider the clustering results in conjunction with the carrying capacity of the robot. If the number of tasks in the clusters does not align with the processing capacity of the robot, it will result in the robot needing to incur a larger movement cost, thereby reducing the robot’s efficiency in completing the tasks. Sumana [23] proposed a task assignment method based on nearest neighbor search and integrated it with path planning for multi-intelligent agents to effectively address the task assignment problem in dynamic environments. The task is assigned to neighboring multi-intelligent agents after clustering using the K-means algorithm and integrating it with the path planning of the Particle Swarm Optimization (PSO) algorithm. Simulation experiments demonstrate that, using this method, the multi-intelligent agents can complete the assigned tasks in a complex environment. However, the method does not take into consideration the clustering results and the carrying capacity of the robots. Table 1 summarizes the future directions and trends in the development of commonly used algorithms for multi-robot task allocation.

This paper proposes a task allocation method based on the synergy of clustering and heuristic intelligent algorithms. The K-means++ clustering algorithm is used to divide the total task set into multiple disjoint clustered tasks sets to reduce the individual dimension of the intelligent algorithm. At the same time, a condition limiting the maximum number of clusters is added to the K-means++ clustering algorithm to ensure that the clustering result matches the robots’ processing capability. Then, a fitness function is established, and the clustering result is reasonably allocated to each robot by the PSO algorithm. Finally, the ordering of each clustered task set is optimized by the PSO algorithm so that each robot obtains an optimally ordered clustered task set. The results of simulation experiments, conducted using the Robot Operating System (ROS), as well as real robot experiments, demonstrate that the algorithm proposed in this paper surpasses other comparative algorithms in terms of task assignment time, total distance traveled to complete the task, and overall time to complete the task.

The rest of the paper is organized as follows: Section 2 describes the mathematical model for multi-robot task allocation and the framework design of the multi-robot system. Section 3 highlights the detailed process of the multi-robot task allocation algorithm. Section 4 and Section 5 present the simulation experimental study and the real robot experimental study, respectively, and compare and analyze the algorithm proposed in this paper with other algorithms. Section 6 and Section 7 consist of the discussion and conclusion sections of the article.

## 2. A Mathematical Model and Systematic Framework for Multi-Robot Task Allocation

Establishing a mathematical model for multi-robot task allocation and designing a system framework for the same purpose are crucial for ensuring the efficient operation and collaborative work of multi-robot systems. These endeavors can achieve optimal allocation of robot resources, enhance the efficiency of task execution, and foster collaboration and coordination among robots through precise mathematical models and a well-designed system framework.

Suppose there are m robots in a room with n task points, and the position of each robot and the position of the task points are known, now it is necessary to make these m robots return to their respective initial positions after performing n tasks. Each task can be performed by only one robot.

### 2.1. Mathematical Model of Multi-Robot Task Allocation

The variables in the multi-robot task assignment system of this paper are defined as follows:

The set of robots is *R* = {*R*_1_, *R*_2_, *R*_3_, …*R_i_*, …*R_m_*}, where *m* denotes the number of the robots, and *R_i_* denotes robot *i*. The set of tasks is *T* = {*T*_1_, *T*_2_, *T*_3_, …*T_k_*, …*T_n_*}, where *n* denotes the total number of tasks in the system and *T_k_* denotes the *k*th task. The set of tasks is clustered according to the number of robots and divided into a set of *m* disjoint clusters equal to the number of robots, *C* = {*C*_1_, *C*_2_, *C*_3_, …*C_j_*, …*C_m_*}, where *C_j_* denotes the *j*th cluster and the following conditions are satisfied: *T* = *C*_1_∪*C*_2_∪…*C_l_*…∪*C_m_* (all clusters of the concatenation set are equal to the original task set *T*), and for any *j* ≠ *l*, there is *C_j_*∩*C_l_* = ø (no intersection between any two clusters), where *C_j_* = {$T_{1}^{j}$, $T_{2}^{j}$, …$T_{l.}^{j}$, …$T_{nl}^{j}$}, $T_{l}^{j}$ denotes the *l*th task point in the cluster. *L* Є {1, 2, …, *n_l_*}, *n_l_* is the number of tasks in *C_j_*, *n*_1_ + *n*_2_ + … + *n_l_* + … + *n_m_* = *n* where *n* is the total number of tasks in the set of tasks *T*.

The multi-robot task allocation is based on the shortest total distance traveled by multiple robots as the optimization objective function, and the mathematical model is established as:(1)$$minCost_{1}=\sum_{i=1}^{m}\sum_{j=1}^{m}x_{ij}d\left(R_{i},C_{j}\right)$$
(2)$$minCost_{2}=\sum_{i=1}^{m}\sum_{k=1}^{n_{m}}d(R_{i},T_{n_{m}}^{m})x_{ik}$$
(3)$$s.t\sum_{k=1}^{n_{m}}x_{ik}=1(i=1,2\ldots m)$$
(4)$$\sum_{i=1}^{m}x_{ik}=1(k=1,2\ldots n_{m})$$

Equation (1) is the first objective function 1, which indicates that the robot is assigned to the cluster with the shortest distance from itself, using the shortest distance as the evaluation index. Here, *x_ij_* is the decision variable between the robot and the cluster center, which indicates whether the robot and the cluster center are accepted or not; *x_ij_* = 1 indicates that the robot accepts the task of the cluster center, otherwise *x_ij_* = 0. The Euclidean distance between robot *R_i_* and cluster center *C_j_* is represented by *d* (*R_i_*, *Cj*)

Equation (2) is the second objective function that represents the sum of the shortest distances for multiple robots to return to their respective initial positions after performing their respective cluster tasks. Here, *d* (*R_i_*, $T_{n_{m}}^{m}$) signifies the Euclidean distances both between the robots and the assigned task target points within the clusters, and between these target points and the designated goal points; *x_ik_* denotes whether the paths connecting the robots to the target points of the tasks, as well as between the target points and the goal points, are selected or not. If selected, *x_ik_* is set to 1; otherwise, *x_ik_* is set to 0. Here, *m* represents the total number of clusters, while *n_m_* indicates the count of target points, or task goal points, within cluster *C_m_*. Equations (3) and (4) denote that each task is visited by each robot one time and only once.

### 2.2. Multi-Robot Tasking System Framework

The process of multi-robot task allocation encompasses three primary stages: environment map generation, task assignment, and path planning. The SLAM (Simultaneous Localization and Mapping) method is employed to gather sensor data, which is then used to construct the map, group the system-issued task points into clusters, and efficiently distribute these clusters among the multi-robot system. Subsequently, each clustered task set undergoes sorting, and ultimately, multi-robot task allocation is achieved through the implementation of path planning. The systematic framework for multi-robot task allocation is illustrated in Figure 1.

This system utilizes the ROS (Robot Operating System) platform to write the relevant nodes in the multi-robot task assignment module, specifically the robot position release node, task point K-means++ clustering node, clustering-based task assignment node, and task set sorting node. A 2D raster map is constructed utilizing odometer data and LiDAR data through the application of the Gmapping [24] algorithm. The multi-robot task allocation is realized using the move_base function package integrated with ROS to obtain odometer information for multi-robot path planning.

## 3. Improved K-Means++ Clustering Algorithm and Particle Swarm Algorithm Synergistic Approach to Task Allocation

By combining cluster analysis with the PSO [25] algorithm, cluster analysis can help the algorithm quickly locate similar task clusters, thus narrowing the search scope and improving search efficiency. On the other hand, the PSO algorithm can further search for the optimal task allocation scheme within these clusters to ensure the rationality and effectiveness of task allocation.

### 3.1. Improved K-Means++ Clustering Algorithm

In the standard K-means [26] algorithm, the initial clustering centers are chosen randomly, which may lead to unstable results and sometimes even result in falling into local optimal solutions. The K-means++ algorithm, on the other hand, selects the initial clustering centers in a more intelligent way to improve the effectiveness and stability of clustering. Although the K-means++ [27] algorithm has high clustering effectiveness and stability, there is no inherent limit to the number of tasks per cluster it can generate. Due to the limited carrying capacity of the robot, which allows it to handle a maximum of six tasks simultaneously, it becomes necessary to ensure that a cluster assigned to the robot does not contain seven tasks, as the robot will not be able to complete them all at once. Consequently, clustering must be constrained in such a way that the maximum number of tasks within any given cluster does not exceed the robot’s capacity. This ensures that the clustering outcomes align with the robot’s carrying capacity, making the results more practical. In this study, we enhance the K-means++ algorithm by introducing a cap on the maximum number of tasks per cluster. The procedure for the enhanced K-means++ algorithm is outlined as follows:

Step 1: Initialization: A task point is arbitrarily chosen from the task set to serve as the initial clustering center.

Step 2: Distance Calculation: For every task point in the set, compute its distance to the chosen clustering center. The formula for this distance between a clustering center and a task point is provided in Equation (5).
(5)$$D(C_{m},T_{n})=\sqrt{{\left(x_{Cm}-x_{Tn}\right)}^{2}+{\left(y_{Cm}-y_{Tn}\right)}^{2}}$$

Equation (5) where *C_m_* is the *m*th clustering center and *T_n_* is the *n*th task point.

Step 3: Determination of Clustering Centers: Based on the computed distance probability distribution, task points that are farther away are selected as subsequent clustering centers with a higher likelihood. This process is repeated until a total of *m* clustering centers have been chosen, as outlined in Equation (6).
(6)$$p=\frac{D^{2}\left(C_{m},T_{n}\right)}{\sum_{m=1}^{m}\sum_{n=1}^{n}D^{2}\left(C_{m},T_{n}\right)}$$

Equation (6) defines the probability *p* of a task point being chosen as the next clustering center.

Step 4: Assign Clusters: Each task point in the set is assigned to the cluster whose center is closest to it.

Step 5: Verify if the task points allocated to each cluster have reached the maximum limit. If not, proceed to the next step. If the maximum limit is reached, distribute the surplus task points to the nearest cluster to the current cluster.

Step 6: Update Cluster Centers: Compute the mean of all task points in each cluster to determine the new cluster center.

Step 7: Iterate: Repeat Steps 4, 5, and 6 until the cluster centers remain unchanged.

### 3.2. Particle Swarm Algorithm for Multi-Robot Task Allocation Based on Clustering Results

Based on the clustering results and the locations of multiple robots, each cluster is reasonably assigned to the multi-robot system using the PSO algorithm. The Particle Swarm Optimization (PSO) algorithm is an optimization technique that mimics the collective behavior of groups, such as bird flocks or fish schools. In the context of multi-robot task allocation, each particle embodies a specific task allocation plan. The particle’s position vector signifies how tasks are assigned to robots, while its velocity vector indicates the direction of adjustment in task assignment. The algorithm assesses the efficacy of various task allocation schemes by computing a fitness function. Particles adjust their velocities and positions based on personal bests and global bests, iteratively converging towards the optimal task allocation scheme. The detailed implementation steps are outlined below:

Step 1: Initialization involves setting up the particle swarm by determining the random initial position and velocity of each particle, and configuring the algorithm parameters ω, c_1_, c_2_.

Step 2: Determine the fitness value by establishing a fitness function based on the clustering center and the robot’s position. The fitness function is Equation (1).

Step 3: Update Individual Particle Optimal Values: For each particle, compare the fitness value of its current position with that of its historical optimal position. If the current position yields a higher fitness value than the previous personal best, update the historical optimal position to the current position.

Step 4: Update Global Best Value: Evaluate the fitness values of all particles’ current positions and compare them with the current global best value. If any particle’s current position has a higher fitness value than the previously recorded global best, update the global optimal position to that particle’s current position.

Step 5: Adjust Particle Position and Velocity: Update the velocity and position of each particle using Equations (7) and (8), respectively.
(7)$$v_{ij}(t+1)=\omega v_{ij}(t)+c_{1}\;r_{1}(p_{ij}(t)\;-x_{ij}(t))+c_{2}\;r_{2}(p_{gj}(t)\;-x_{ij}(t))$$

Equation (7) where: *ω* is the inertia weight coefficient; *r*_1_, *r*_2_ are random numbers between [0,1]; *c*_1_, *c*_2_ are the learning factors. In the *t*th iteration, *v_ij_*(*t* + 1) denotes the velocity of the ith particle in the *j*th dimension, where *j* represents the *D*-dimensional search space. The individual optimal value for the *i*th particle in the *t*th iteration is represented by *p_ij_*(*t*). Meanwhile, *p_gj_*(*t*) signifies the global optimal value across the population in the *t*th iteration.
(8)$$x_{ik}(t+1)=x_{ik}(t)+v_{ik}(t+1)\;,k=1,2,\ldots ,D$$

In Equation (8) *x_ij_*(*t* + 1) represents the position of the *i*th particle in the *j*th dimension during the *t*th iteration.

Step 6: Termination Criterion: The algorithm concludes if the termination condition is met, with the global best position being declared as the optimal solution. If the termination condition is not yet satisfied, the algorithm proceeds to repeat step 2.

### 3.3. Particle Swarm Algorithm to Optimize Clustering Task Set Ordering

The result of the clustering-based PSO algorithm for multi-robot task assignment converts the multi-robot task assignment problem into a multi-traveler problem. Luo [28] executed the tasks in a cluster according to the serial number of the task set within that cluster. Although this method is simpler to execute, it may result in robots needing to pay a large path cost. Therefore, this paper, based on the task allocation results of the PSO algorithm, optimizes the task set ordering within the clusters of task sets. The purpose of this ordering is to find, for each robot, the shortest path from its starting position to the completion of the tasks in its respective cluster. Equation (2) presents the objective function for optimizing the ordering of the task set within clusters. The sorting process for the task set is illustrated in Figure 2.

## 4. Simulation Experiment

### 4.1. Scene Description

In this paper, we simulate the smart workshop environment as the application scenario, and the layout sketch is shown in Figure 3. There are *m* isomorphic mobile robots in the workshop that need to visit *n* production equipment stations, and the products processed by the production equipment are carried to the designated shelves. Gerkey and Mataric [29] categorize the multi-robot task assignment problem based on three dimensions: robot type, task type, and assignment type, specifically into the following categories:(1)Robot type: A single-tasking robot (ST) is one that can only perform one task at a time; a multi-tasking robot (MT) is one that can perform multiple tasks simultaneously.(2)Task types: Single-robot tasks (SR) are those that require only one robot to complete; multi-robot tasks (MR) are those that require multiple robots to complete.(3)Assignment types: Instantaneous assignment (IA) refers to a situation where each robot is assigned one task without future planning; time-expanded assignment (TA) refers to a scenario where a series of tasks can be assigned to a robot within the planning horizon.

Therefore, based on the classifications along the above three dimensions, the robot task assignment problems can be categorized into eight distinct types: ST-SR-IA, ST-SR-TA, MT-SR-IA, MT-SR-TA, ST-MR-IA, ST-MR-TA, MT-MR-IA, and MT-MR-TA. Note that in the context of this paper, the multi-robot task assignment problem belongs specifically to the multi-tasking single-robot time-expanded assignment (MT-SR-TA) category, meaning that each robot is capable of performing multiple tasks simultaneously, but each task within this category is still assumed to be completed by only one robot (despite being multi-tasking, the robot does not collaborate with others on a single task).

The subsequent fundamental assumptions are established for the allocation of tasks among multiple robots:(1)Each robot has a maximum load limit, the transportation process occurs at a uniform speed, and transportation cannot exceed the load limit and is assumed to be in good condition.(2)Adequate power is possessed by each robot to complete all assigned tasks.(3)The robots begin at their designated starting points, travel to the task location to fulfill the transportation tasks, and then return to their initial positions upon completing all assigned tasks.

### 4.2. Simulation Experiment Platform Construction

The simulation experimental platform utilizes the ROS Noetic version and Ubuntu 20.04 system, with laptop parameters of an I5-1155G7 processor at 2.50 GHz and 16 GB of RAM. The layout of the simulation environment follows the design of the smart shop floor depicted in Figure 3, which, as depicted in Figure 4, has a map size of 40 × 35 m. There are three isomorphic mobile robots and 20 task points, along with shelves. The robot coordinates (in meters) are *R*_1_ (5, 0.5), *R*_2_ (13, 0.5), *R*_3_ (21, 0.5); the position of the shelves (in meters) is (28, 2); and the coordinates of the task points are presented in Table 2.

Let the system assign 10 tasks, *T*_1_, *T*_3_, *T*_7_, *T*_8_, *T*_9_, *T*_12_, *T*_14_, *T*_16_, *T*_17_, and *T*_20_, and have 3 robots carry these 10 tasks to the shelves and then return to their respective initial positions. Each robot has the capability to carry out a maximum of 6 handling tasks simultaneously. The K-means++ algorithm is applied to cluster the 10 task points, with a restriction on the maximum number of clusters set at 6. The clustering results are shown in Figure 5.

According to the position of the robots and the results of clustering, the PSO algorithm is used for task allocation, and the allocation results are shown in Figure 6.

As shown in Figure 6 it is observed that Robot1, Robot2, and Robot3 are assigned to the clustering centers 1, 2, and 3, respectively, to perform the tasks in their respective clusters. After assigning the clustered tasks to multiple robots, the PSO algorithm is then used to sort the task set, and the sorting results are shown in Figure 7.

The simulation process of the three robots performing the tasks in their respective clusters is shown in Figure 8. In the diagram, the green line denotes the path of Robot1, the red line shows the path of Robot2, and the yellow line illustrates the path of Robot3. Each robot travels at a speed of 0.5 m per second. The three robots start from the initial position in turn, complete all their respective handling tasks, and then return to their initial positions. Among the tasks, generating the robot’s moving trajectory is a complex and crucial process that involves path planning and motion control. The path planning and motion control for the robot, as used in this paper, are integrated within the move_base function package of ROS. Path planning involves searching for the optimal path from the starting point to the target point on the constructed map, using a path-planning algorithm. This algorithm usually takes into account the length of the path, obstacles, and other relevant information to find the best solution. Motion control is the process of controlling the robot’s mobile chassis to follow the planned path, resulting in the generation of a moving trajectory.

### 4.3. Comparison of Simulation Experiments

To verify the impact of the algorithm used in this paper (K-means++ + PSO) on the efficiency of multi-robot collaborative processing, it is compared with the clustering-based market auction algorithm (K-means + MAA) [17], the non-clustered particle swarm algorithm (PSO) and the clustering-based genetic algorithm (K-means + GA) [22]. Let the system generate 14 tasks, namely *T*_1_, *T*_3_, *T*_4_, *T*_7_, *T*_8_, *T*_9_, *T*_11_, *T*_12_, *T*_13_, *T*_14_, *T*_16_, *T*_17_, *T*_19_, and *T*_20_. The comparative results of the task assignments among the four algorithms in the simulation experiments are illustrated in Figure 9. Among them, Figure 9a,c,e,g shows the task set ordering diagrams of the four robots performing their respective tasks using the K-means++ + PSO algorithm, K-means + MAA algorithm, PSO algorithm, and K-means + GA algorithm, respectively. Figure 9b,d,f,h shows the trajectory diagrams of the four robots performing their respective tasks under the three algorithms, respectively. In these diagrams, the green color represents the moving trajectory of robot 1, the red color represents the moving trajectory of robot 2, and yellow represents the moving trajectory of robot 3.

To minimize the error, each of the four algorithms performs five experiments, with a different number of tasks in each experiment. The simulation results for multi-robot task assignment time, total multi-robot task completion time, and total distance completed by multi-robot task assignment are obtained, as shown in Figure 10, Figure 11 and Figure 12.

### 4.4. Simulation Experiment Results and Analysis

As can be seen from Figure 9, the K-means++ + PSO algorithm limits the maximum number of tasks in the clusters according to the carrying capacity of the robots, so the task allocation in Figure 9b is more reasonable, and the three robots can carry their respective assigned tasks at one time. From Figure 9d,h, it can be seen that in the K-means + MAA algorithm and K-means + GA algorithm, because the maximum number of tasks in the clusters has not been limited, Robot3 has been assigned seven tasks and eight tasks, so Robot3 cannot carry its tasks to the shelves at one time and needs to carry them twice. From Figure 9f, Robot2 using the PSO algorithm also needs to carry tasks twice to complete the carrying task, which increases the energy consumption and time cost of the robots. From Figure 10, it can be seen that the K-means++ + PSO algorithm shortens the task allocation time by 68.05% compared to the PSO algorithm, 49.84% compared to the K-means + MAA algorithm, and 33.68% compared to the K-means + GA algorithm. From Figure 11, it can be seen that the K-means++ + PSO algorithm reduces the time consumed to complete tasks by 5.26% compared to the K-means + MAA algorithm, by 4.0% compared to the K-means + GA algorithm, and by 12.75% compared to the PSO algorithm. Figure 12 shows that the K-means++ + PSO algorithm shortens the total distance traveled to complete the assigned tasks by 5.31% compared with the K-means + MAA algorithm, by 4.58% compared with the K-means + GA algorithm, and by 12.30% compared with the PSO algorithm. The algorithm proposed in this paper effectively improves the efficiency of the task assignment of multiple robots in the intelligent workshop.

## 5. Real Robot Experiments

### 5.1. Real Robot Experiments Platform Construction

The multi-robot utilized in the real robot experiments is depicted in Figure 13. The hardware includes the main control board (Raspberry Pi 4B), LIDAR (YDLIDAR X2L), motor drive module, inertial measurement unit (IMU), two-wheel differential chassis, laptop, and router.

An artificial experimental setting has been constructed, depicted in Figure 14. There are three isomorphic mobile robots with 20 task points and shelves.

The Gmapping algorithm is employed to construct SLAM maps of the experimental environment, as illustrated in Figure 15. The Gmapping algorithm is based on a particle filter, a probabilistic filter utilizing Monte Carlo methods. It estimates the robot’s state by maintaining a set of particles, each representing a possible position of the robot and the corresponding map feature. During the robot’s movements, the Gmapping algorithm updates the weights of the particles based on the environmental information scanned by the LiDAR and the robot’s movements, thereby enabling the estimation of the robot’s position and the construction of the map [24]. The coordinate positions of 20 task points and a shelf are recorded according to the odometer information during the map building process. The shelf position (in meters) is (−0.3, 6) and the coordinate position details for the 20 task points are presented in Table 3.

### 5.2. Real Robot Experiments Process

Let the system release 10 tasks. Three robots are needed to carry these 10 tasks to the shelves and then return to their initial positions. The robots’ coordinates are as follows: *R*_1_ (0.5, −0.5), *R*_2_ (0.5, 1.0), and *R*_3_ (0.5, 2.5). The 10 tasks are *T*_1_, *T*_3_, *T*_7_, *T*_8_, *T*_9_, *T*_12_, *T*_14_, *T*_16_, *T*_17_, and *T*_20_. Each robot can perform up to 6 handling tasks at a time.

The clustering of the 10 task points uses the K-means++ algorithm, with the maximum number of task points in each cluster limited to 6. The clustering results are as follows: cluster 1 includes {*T*_1_, *T*_3_, *T*_7_, *T*_8_}, cluster 2 includes {*T*_12_, *T*_14_, *T*_20_}, and cluster 3 includes {*T*_9_, *T*_16_, *T*_17_}. Then the PSO algorithm is used for task assignment based on the location of the center of each cluster and the location of the three robots. The results of the task assignment are the following: robot 1 is assigned to cluster 1, robot 2 is assigned to cluster 2, and robot 3 is assigned to cluster 3 to perform their respective tasks. Finally, the tasks in each cluster are then sorted using the PSO algorithm for the task set, and the sorted results are the following: robot 1 → *T*_3_ → *T*_1_ → *T*_8_ → *T*_7_ → shelf → robot 1, robot 2 → *T*_12_ → *T*_14_ → *T*_20_ → shelf → robot 2, robot 3 → *T*_9_ → *T*_16_ → *T*_17_ → *T*_20_ → shelf → robot 3.

The experimental outcomes for the multi-robot task allocation using the algorithm introduced in this paper are displayed in Figure 16. In Figure 16a three robots go to execute their first tasks: robot 1 goes to execute task *T*_3_, robot 2 goes to execute task *T*_12_, and robot 3 goes to execute task *T*_9_. Figure 16b shows that robot 2 goes to execute task *T*_14_ after executing task *T*_12_. According to Figure 16c, Robot 2 proceeds to perform task *T*_20_ following the completion of task *T*_14_, while Robot 1 moves on to perform task *T*_1_ after completing task *T*_3_. Figure 16d indicates that robot 2 proceeds towards the shelf upon completing task *T*_20_, robot 1 transitions to task *T*_8_ following the completion of *T*_1_, and robot 3 undertakes task *T*_16_ after finishing *T*_9_. Figure 16e shows that robot 2 moves from the shelf position to the initial position after performing all handling tasks; robot 1 moves to the shelf position after performing task *T*_7_, and robot 3 moves to the shelf position after performing task *T*_17_. Figure 16f demonstrates that the three robots revert to their original positions once they complete their individual tasks.

### 5.3. Comparison of Real Robot Experiments

Figure 17 shows the experimental comparison between the K-means++ + PSO algorithm, K-means + MAA algorithm, PSO algorithm, and K-means + GA algorithm for three robots after performing their respective tasks. Figure 17a,c,e,g shows the task set ordering diagrams of the three robots after performing their respective tasks using the K-means++ + PSO algorithm, K-means + MAA algorithm, PSO algorithm, and K-means + GA algorithm, respectively. Figure 17b,d,f,h shows the trajectory diagrams of the three robots after performing their respective tasks under the four algorithms, with green representing the trajectory of Robot 1, red representing the trajectory of Robot 2, and yellow representing the trajectory of Robot 3, respectively.

### 5.4. Real Robot Experiments Results and Analysis

Five experiments were conducted for each of the four algorithms, each with a different number of tasks. As can be seen in Table 4, our proposed K-means++ + PSO algorithm reduced the task allocation time by 51.25% compared to the K-means + MAA algorithm, 68.82% compared to the non-clustered PSO algorithm, and 46.24% compared to the K-means + GA algorithm. The total distance traveled to complete the assigned task was also reduced by 6.06%, 10.04%, and 5.97%, respectively. The time consumed to complete the task was reduced by 5.12%, 9.95%, and 4.0%, respectively. The experimental results show that the algorithm proposed in this paper improves the efficiency of multi-robot material handling in an intelligent workshop.

## 6. Discussion

Through simulation experiments and real robot experiments, the algorithm proposed in this paper shows good performance. Firstly, according to the carrying capacity of the robot, the limitation of the maximum number of clusters is added to K-means++, so that the clustering results match with the carrying capacity of the robot, which indicates that the robot only needs to complete all the assigned tasks at one time. In addition, K-means++ compared with the K-means algorithm shows that the K-means++ algorithm is more optimized in the method of initializing the clustering centroids, which leads to better clustering results. Therefore, the K-means++ algorithm has a strong advantage over the K-means algorithm, which does not limit the maximum number of clusters, to reduce the cost of robot movement. Secondly, assigning the clustering results to multiple robots using the PSO algorithm reduces the complexity of task assignment while increasing the accuracy of task assignment compared to task assignment using a non-clustered particle swarm algorithm. In the figure it can be seen that with the non-clustered particle swarm algorithm, the task assignment is less effective resulting in a larger path cost for the robots. Finally, the particle swarm algorithm sorts each clustered task set so that each robot performs the task at the minimum cost of movement, which further improves the efficiency of multi-robot cooperative work.

Although the algorithm proposed in this paper has certain advantages in multi-robot task allocation, there are some limitations to the method, as follows: (1) this paper does not consider the condition of task priority, which may pose challenges for the algorithm in practical applications if task priorities change; (2) this paper does not account for robot faults, and the algorithm will be unable to address the task allocation problem when robots are faulty.

In the future, we hope to incorporate task priority and dynamic change factors into the algorithm design, so that the algorithm can adjust the task allocation scheme according to actual situations, thereby improving the adaptability and robustness of the algorithm. In addition, we also aim to extend this task allocation algorithm to other fields, such as healthcare and agriculture, where its effectiveness and reliability can be validated through actual data.

## 7. Conclusions

In this study, addressing the challenge of multi-robot task assignment through clustering, we introduce a collaborative intelligent approach that integrates clustering techniques with the particle swarm optimization (PSO) algorithm. This method clusters the task set using the K-means++ algorithm, incorporating the consideration of the robot’s processing capacity and setting a limiting condition on the maximum number of clusters to make the clustering results more reasonable. Then, the clustering results are reasonably assigned to multiple robots using the PSO algorithm. Additionally, the order in which robots execute their respective clustered task sets is optimized by the PSO algorithm, ensuring that each robot obtains the shortest execution path. Simultaneously, an intelligent workshop environment is simulated to create both simulation and physical scenes for conducting comparative experiments with different algorithms. The experimental results demonstrate that the task allocation of the algorithm proposed in this paper is more reasonable, as it reduces the time and moving distance required for executing tasks, ultimately improving the efficiency of multiple robots’ work.

## Figures and Tables

**Figure 1 biomimetics-09-00694-f001:**
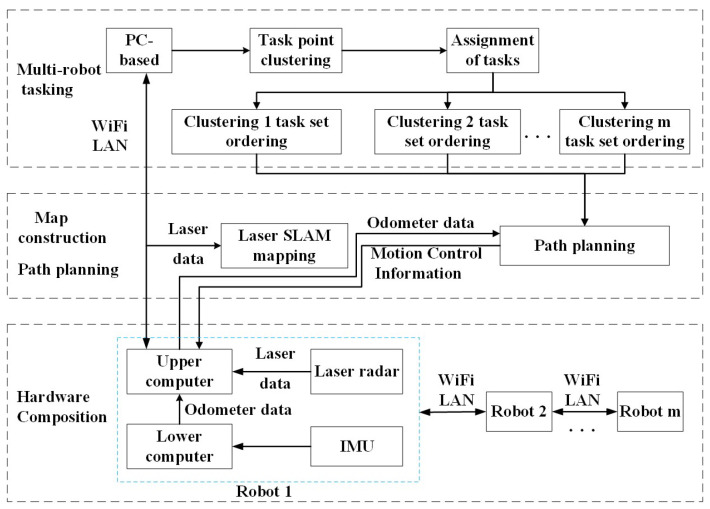
Framework diagram of multi-robot task allocation system.

**Figure 2 biomimetics-09-00694-f002:**
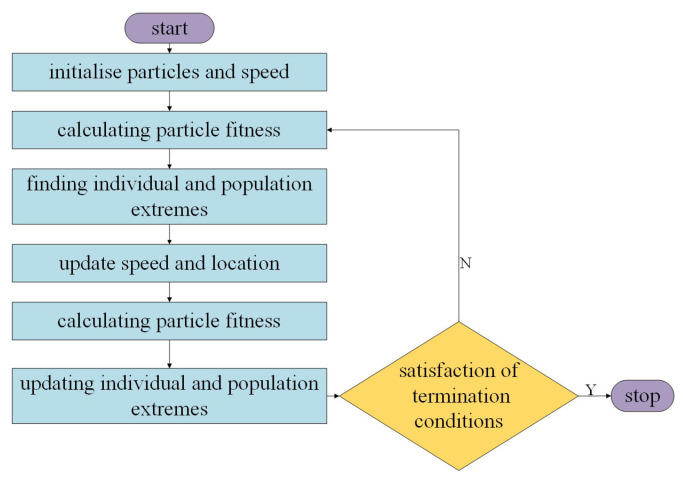
Clustering task set sorting flowchart.

**Figure 3 biomimetics-09-00694-f003:**
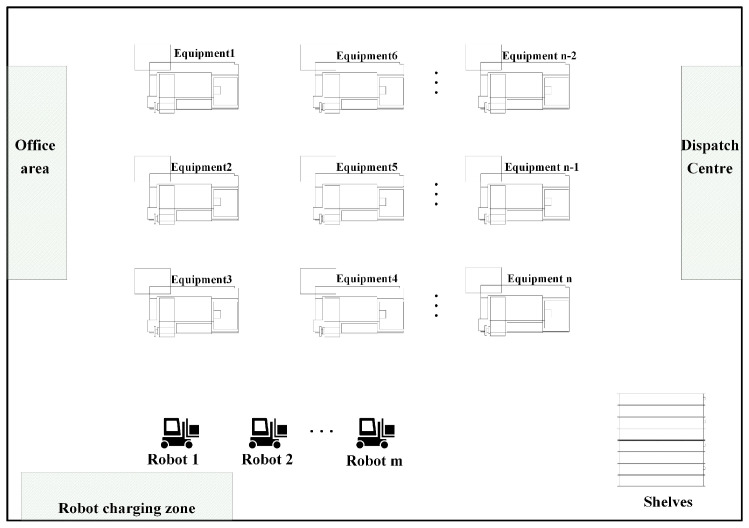
Layout sketch of intelligent workshop.

**Figure 4 biomimetics-09-00694-f004:**
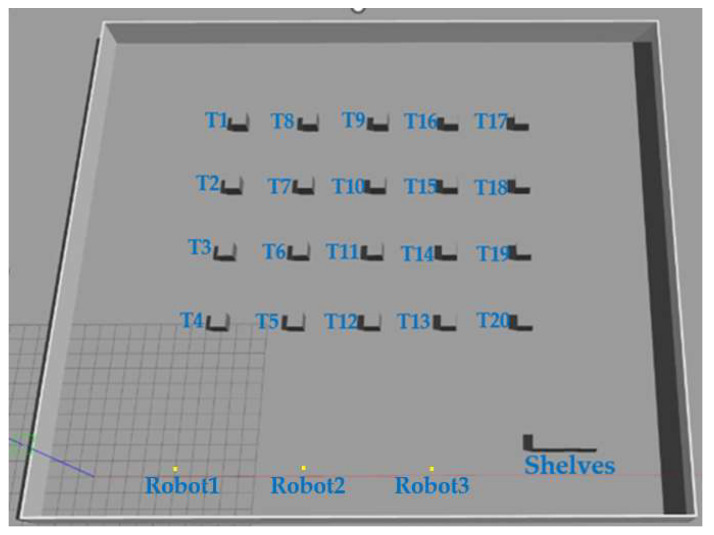
Gazebo Simulation environment.

**Figure 5 biomimetics-09-00694-f005:**
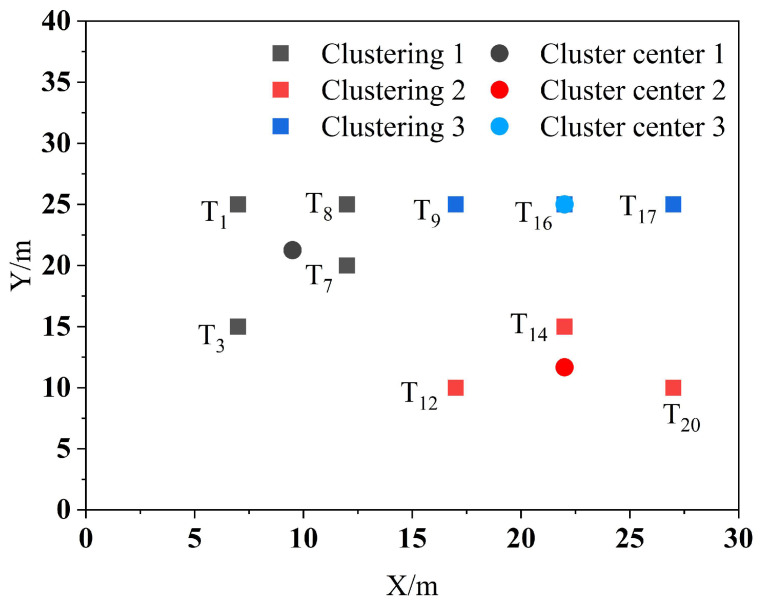
Task point clustering results.

**Figure 6 biomimetics-09-00694-f006:**
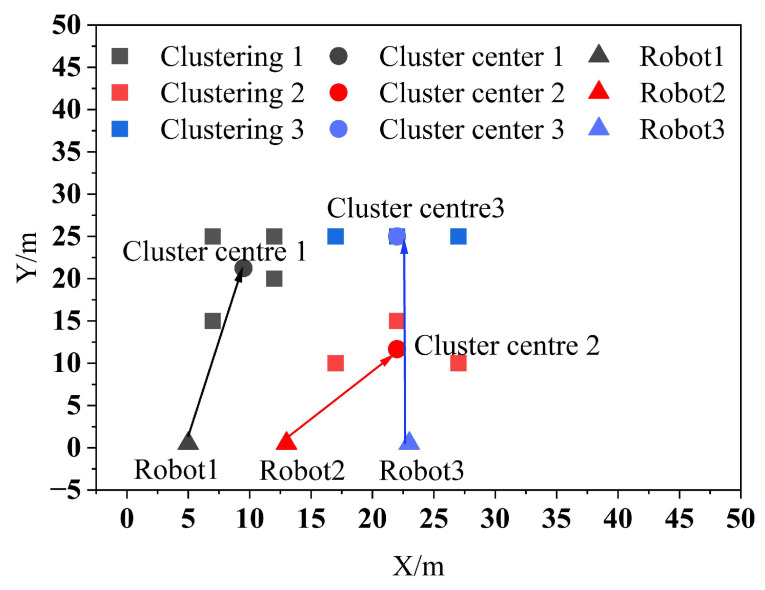
Multi-robot task allocation results.

**Figure 7 biomimetics-09-00694-f007:**
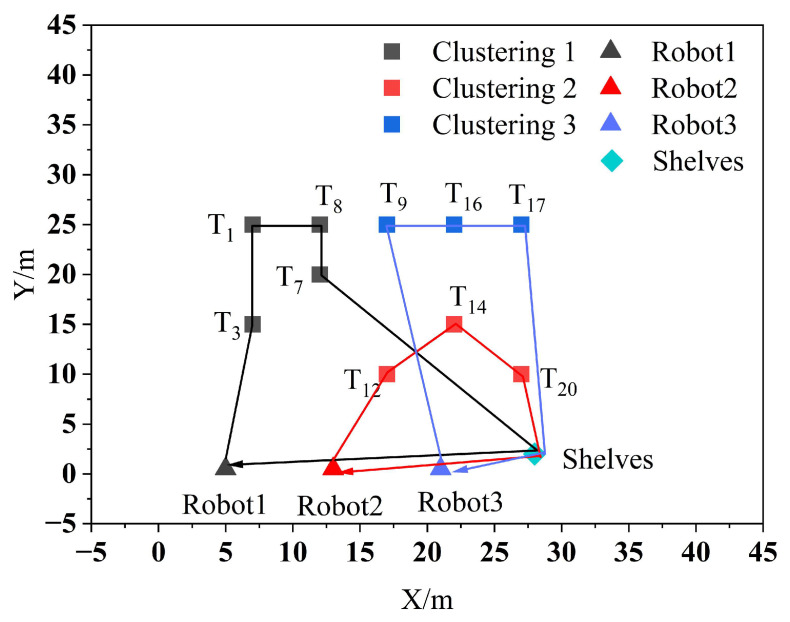
Clustering task set sorting results.

**Figure 8 biomimetics-09-00694-f008:**
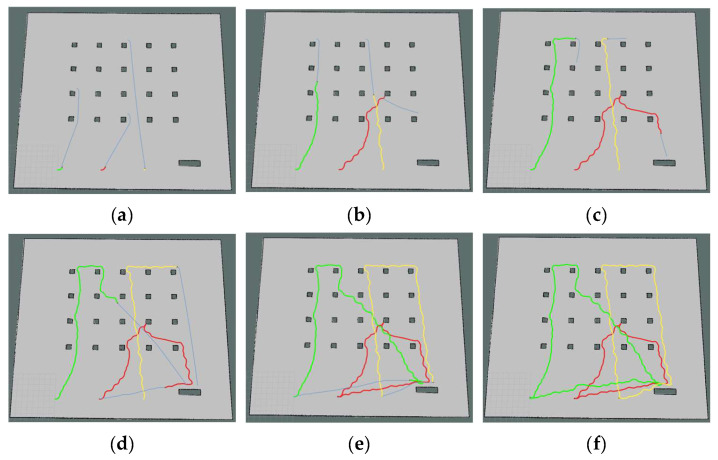
Multi-robot task allocation simulation implementation process. (**a**–**f**) Trajectory diagrams of the three robots performing their respective tasks.

**Figure 9 biomimetics-09-00694-f009:**
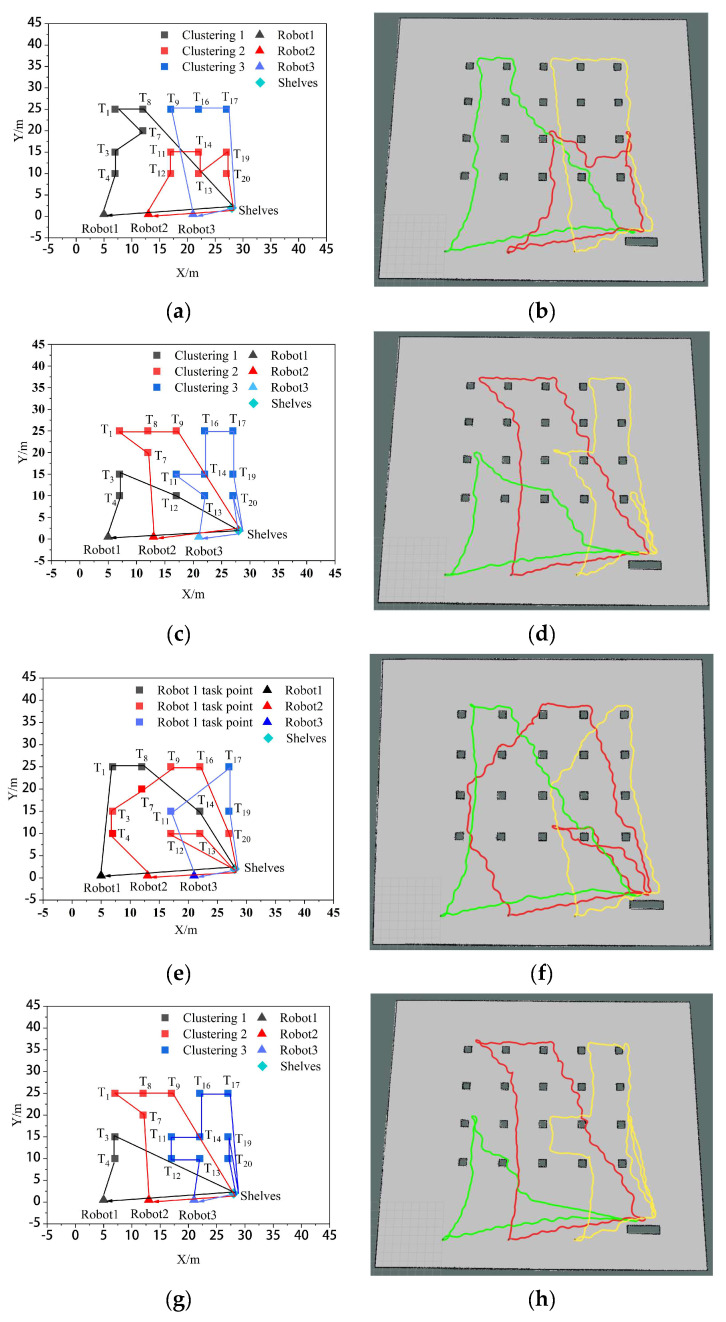
Multi-robot task allocation simulation comparison experiment. (**a**,**c**,**e**,**g**) Sorting plots of the set of tasks after assignment by the four algorithms,. (**b**,**d**,**f**,**h**) Plot of the trajectories of the three robots performing their respective tasks for the four algorithms.

**Figure 10 biomimetics-09-00694-f010:**
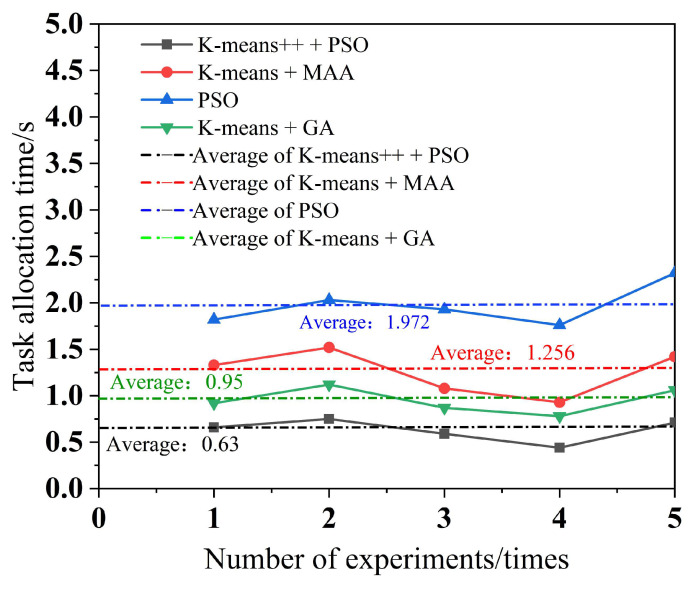
Multi-robot task allocation time.

**Figure 11 biomimetics-09-00694-f011:**
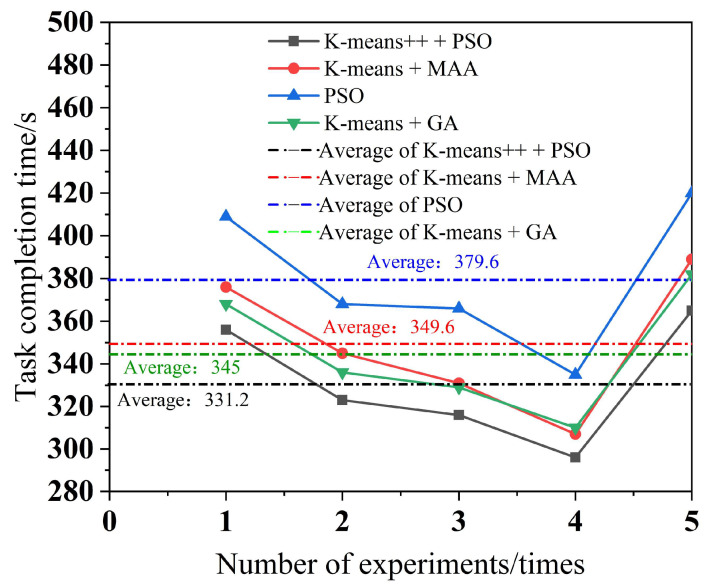
Total multi-robot task completion time.

**Figure 12 biomimetics-09-00694-f012:**
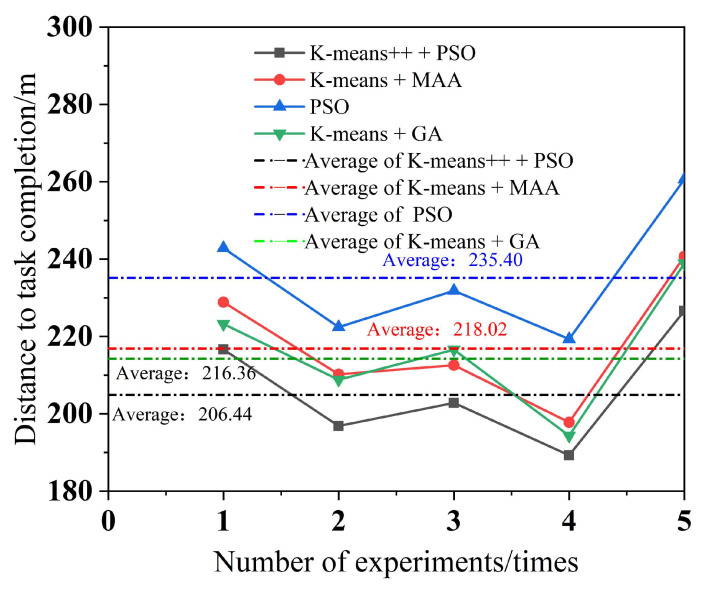
Total distance completed for multi-robot tasking.

**Figure 13 biomimetics-09-00694-f013:**
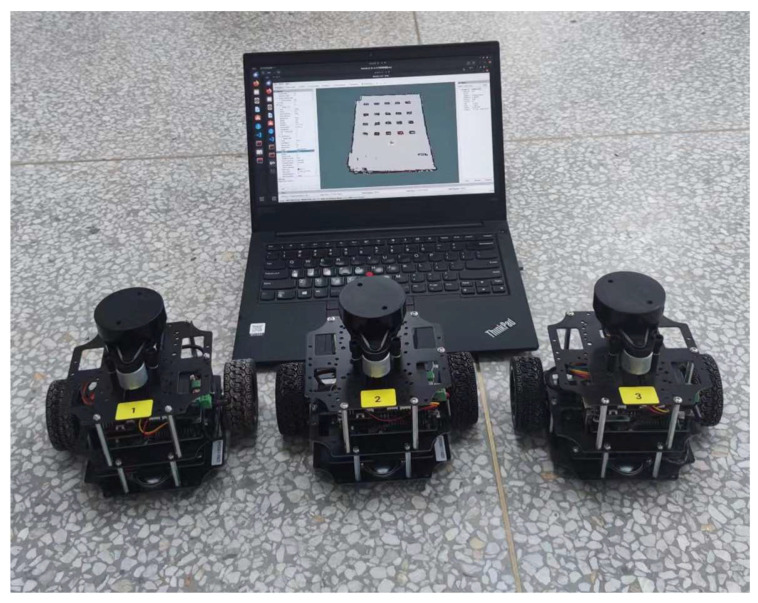
Experimental prototype.

**Figure 14 biomimetics-09-00694-f014:**
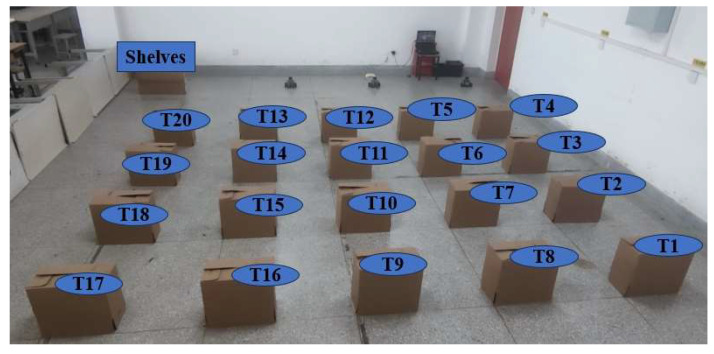
Real robot experiments environment.

**Figure 15 biomimetics-09-00694-f015:**
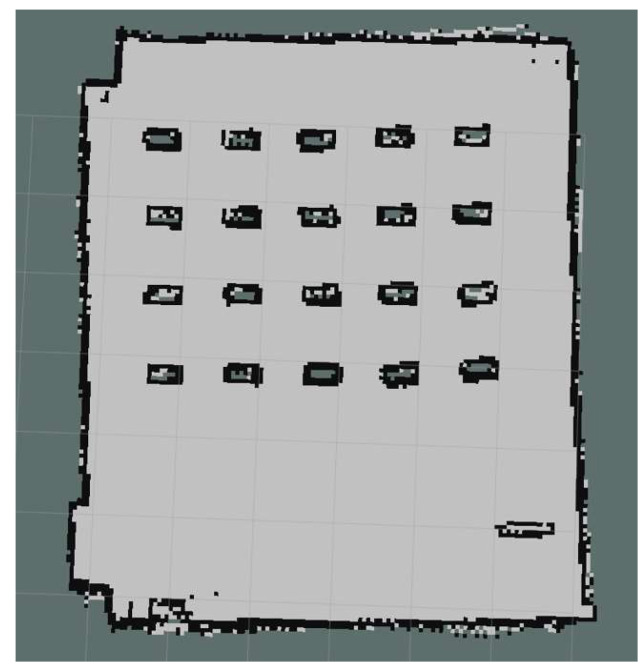
SLAM construction of maps.

**Figure 16 biomimetics-09-00694-f016:**
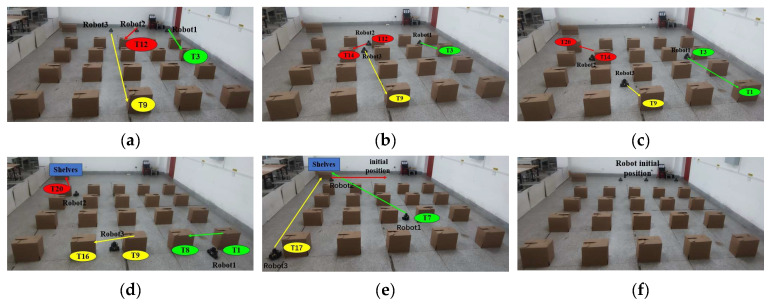
Experimental procedures for multi-robot task assignment with real robots. (**a**–**f**) Diagram of the process of the three robots performing their respective tasks. Where green bubbles represent tasks in cluster 1, red bubbles represent tasks in cluster 2, and yellow bubbles represent tasks in cluster 3.

**Figure 17 biomimetics-09-00694-f017:**
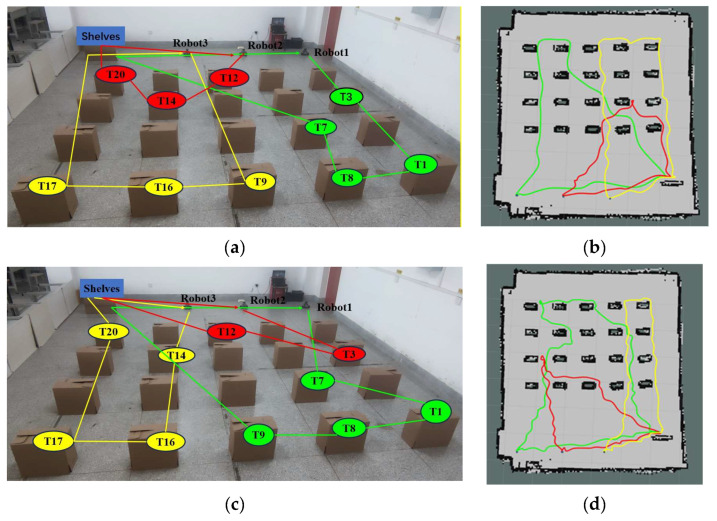
Comparative tests of real robot experiments on multi-robot task assignment. (**a**,**c**,**e**,**g**) Sorting plots of the set of tasks after assignment by the four algorithms. (**b**,**d**,**f**,**h**) Plot of the trajectories of the three robots performing their respective tasks for the four algorithms.

**Table 1 biomimetics-09-00694-t001:** Future directions and trends in the multi-robot task allocation problem.

Task Allocation Optimization Algorithm	Future Research Directions and Trends
Market-based task allocation	1. The construction of a robust and reliable communication network is a necessary condition for market tasking, which has not yet been addressed.
Task allocation based on heuristic algorithms	1. There is a wide variety of heuristic algorithms, each with its own advantages and disadvantages. Future research could explore how to integrate different heuristic algorithms to fully utilize their respective advantages and improve the efficiency and quality of task allocation.
Clustering-based task allocation	1. How to determine the optimal number of tasks in a cluster is a direction for further research. 2. Developing effective switching strategies between clusters to cope with uncertainties such as robot failures is a direction for further research.
Alternative methods of allocating tasks	1. Fully apply artificial intelligence, reinforcement learning and other technologies to optimize task allocation to improve the autonomous decision-making ability and adaptability of the robot. 2. The use of a multiple algorithm fusion strategy for task allocation is also a future research direction

**Table 2 biomimetics-09-00694-t002:** Simulation experiment task information.

Task	Position/m	Task	Position/m	Task	Position/m	Task	Position/m
*T* _1_	(7, 25)	*T* _6_	(12, 15)	*T* _11_	(17, 15)	*T* _16_	(22, 25)
*T* _2_	(7, 20)	*T* _7_	(12, 20)	*T* _12_	(17, 10)	*T* _17_	(27, 25)
*T* _3_	(7, 15)	*T* _8_	(12, 25)	*T* _13_	(22, 10)	*T* _18_	(27, 20)
*T* _4_	(7, 10)	*T* _9_	(17, 25)	*T* _14_	(22, 15)	*T* _19_	(27, 15)
*T* _5_	(12, 10)	*T* _10_	(17, 20)	*T* _15_	(22, 20)	*T* _20_	(27, 10)

**Table 3 biomimetics-09-00694-t003:** Real robot experiments task information.

Task	Position/m	Task	Position/m	Task	Position/m	Task	Position/m
*T* _1_	(−4.8, 0.3)	*T* _6_	(−2.8, 1.3)	*T* _11_	(−2.8, 2.3)	*T* _16_	(−4.8, 3.3)
*T* _2_	(−3.8, 0.3)	*T* _7_	(−3.8, 1.3)	*T* _12_	(−1.8, 2.3)	*T* _17_	(−4.8, 4.3)
*T* _3_	(−2.8, 0.3)	*T* _8_	(−4.8, 1.3)	*T* _13_	(−1.8, 3.3)	*T* _18_	(−3.8, 4.3)
*T* _4_	(−1.8, 0.3)	*T* _9_	(−3.8, 2.3)	*T* _14_	(−2.8, 3.3)	*T* _19_	(−2.8, 4.3)
*T* _5_	(−1.8, 1.3)	*T* _10_	(−2.8, 2.3)	*T* _15_	(−3.8, 3.3)	*T* _20_	(−1.8, 4.3)

**Table 4 biomimetics-09-00694-t004:** Comparison of parameters for running results of multi-robot task assignment.

Algorithms	No. of Experiments/Times	Number of Tasks /Number	Distance to Mission Completion/m		Task Completion Time/s	Task Allocation Time/s
			Three-Robot Travelling Distance d = {d1, d2, d3}/m	Total Distance/m		
K-means++ + PSO	1	8	{19.31, 16.32, 15.05}	50.68	84.90	0.59
	2	10	{19.85, 15.23, 18.67}	53.75	92.35	0.66
	3	13	{20.87, 18.20, 17.46}	56.53	98.62	0.79
	4	15	{25.29, 20.23, 14.86}	60.32	103.21	0.83
	5	16	{28.34, 17.63, 16.21}	62.18	107.25	0.92
Average				56.69	97.266	0.758
K-means + MAA	1	8	{20.56, 13.87, 18.69}	53.12	89.82	1.36
	2	10	{23.45, 15,64, 17.99}	57.08	97.24	1.48
	3	13	{18.32, 25.16, 16.55}	60.03	103.55	1.59
	4	15	{20.85, 26.85, 16.12}	63.82	108.16	1.63
	5	16	{29.68, 19.86, 18.17}	67.71	113.79	1.72
Average				60.35	102.512	1.556
PSO	1	8	{17.13, 22.59, 15.32}	55.04	93.39	1.86
	2	10	{18.45, 25.68, 15.62}	59.39	102.51	2.32
	3	13	{21.29, 29.12, 12.33}	62.74	111.16	2.45
	4	15	{19.37, 30.84, 17.35}	67.56	113.16	2.68
	5	16	{22.56, 32.36, 15.47}	70.39	119.69	2.85
Average				63.02	108.014	2.432
K-means++ + GA	1	8	{19.94, 17.32, 15.80}	52.96	87.35	1.18
	2	10	{24.26, 18.53, 13.88}	56.67	96.67	1.26
	3	13	{26.13, 19.75, 15.47}	61.35	102.34	1.45
	4	15	{26.85, 20.65, 16.12}	63.62	106.56	1.52
	5	16	{31.23, 19.56, 16.08}	66.87	113.68	1.66
Average				60.29	101.32	1.41
K-means++ + PSO reduces % compared to K-means + MAA				6.06	5.12	51.25
K-means++ + PSO reduces % compared to PSO				10.04	9.95	68.82
K-means++ + PSO reduces % compared to K-means + GA				5.97	4.00	46.24

## Data Availability

The data presented in this study are available on request from the corresponding author.

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
