# Peer review of "A Multi-Robot Task Allocation Method Based on the Synergy of the K-Means++ Algorithm and the Particle Swarm Algorithm"

_biomimetics, 2024, doi:10.3390/biomimetics9110694_

Round 1

Reviewer 1 Report

Comments and Suggestions for Authors

An interesting work, describing numerical results and the results of a full-scale experiment with robots.

The main remark concerns the style of presentation of the material.

1. In the final part of the introduction, it would be good to separately state what specific new results the authors claim. Now it is indicated what new things they did, but what are the results?

2. At the beginning of the work, a description of the analyzed model is given in a rather abstract form (section 2.1), from which, frankly speaking, it is unclear (if you do not look at the end of the work) what is being optimized and what problems the robots solve. I think it would be good to essentially state the specific content of the problem being solved before describing the abstract model, what specific actions should be performed by robots. This would significantly facilitate the initial reading of the work.

3. It is necessary to make appropriate edits to the annotation: indicate specific results and detail what was done with robots in the experiment.

4. There is no "Discussion" section, it should usually be there.

5. In general, there are no comments on the paper layout. The pictures are legible, I have no questions about the text.

Author Response

Comments 1: [In the final part of the introduction, it would be good to separately state what specific new results the authors claim. Now it is indicated what new things they did, but what are the results?]

Response 1: We would like to thank the reviewer for this important comment.

We have added new results from the study of this paper in the last part of the introduction of the revised manuscript

The supplementary content is as follows ” [The results of simulation experiments, conducted using the Robot Operating System (ROS), as well as real robot experiments, demonstrate that the algorithm proposed in this paper surpasses other comparative algorithms in terms of task assignment time, total distance traveled to complete the task, and overall time to complete the task.]”

Where in the revised manuscript this change can be found [page 3, and L93-97.]

Comments 2: [At the beginning of the work, a description of the analyzed model is given in a rather abstract form (section 2.1), from which, frankly speaking, it is unclear (if you do not look at the end of the work) what is being optimized and what problems the robots solve. I think it would be good to essentially state the specific content of the problem being solved before describing the abstract model, what specific actions should be performed by robots. This would significantly facilitate the initial reading of the work.]

Response 2: We would like to thank the reviewer for this important comment.

We have added the specific problem being addressed by multi-robotics before describing the task assignment model in Section 2.1 of the revised manuscript

The supplementary content is as follows:” [Establishing a mathematical model for multi-robot task allocation and designing a system framework for the same purpose are crucial for ensuring the efficient operation and collaborative work of multi-robot systems. These endeavors can achieve optimal allocation of robot resources, enhance the efficiency of task execution, and foster collaboration and coordination among robots through precise mathematical models and a well-designed system framework.

Suppose there are m robots in a room with n task points, the position of each robot and the position of the task points are known, now it is necessary to make these m robots return to their respective initial positions after performing n tasks. Each task can be performed by only one robot.]”

Where in the revised manuscript this change can be found [page 3, and L106-115.]

Comments 3: [It is necessary to make appropriate edits to the annotation: indicate specific results and detail what was done with robots in the experiment.]

Response 3: We would like to thank the reviewer for this important comment.

We have edited the annotation appropriately in the revised manuscript

The supplementary content is as follows:” [Song [20] proposed a multi-robot task allocation method based on near-field subset partitioning to solve the problem of inefficiency in the distribution of medical supplies. The algorithm first utilizes the ant colony algorithm [21] to order the task set to form a chain of tasks related to the near field. Then, an objective optimization function is designed based on the task completion time and the path cost of the robots, and a genetic algorithm is used to divide the subsets of this task chain. Then, the divided subset of tasks is assigned to each robot. Finally, the effectiveness of the algorithm is verified by simulating an application scenario in a hospital ward.

Janati [22] first used k-means to group the tasks and then allocate them. Subsequently, they used the Genetic Algorithm (GA) to optimize the clustering results. Through simulation experiments, it can be observed that this method can effectively handle a large number of tasks and address the task allocation problem for robots. However, this method does not consider the clustering results in conjunction with the carrying capacity of the robot. If the number of tasks in the clusters does not align with the processing capacity of the robot, it will result in the robot needing to incur a larger movement cost, thereby reducing the robot's efficiency in completing the tasks.

Sumana [23] proposed a task assignment method based on nearest neighbor search and integrated it with path planning for multi-intelligent agents to effectively address the task assignment problem in dynamic environments. The task is assigned to neighboring multi-intelligent agents after clustering using the k-means algorithm and integrating it with the path planning of the Particle Swarm Optimization (PSO) algorithm. Simulation experiments demonstrate that, using this method, the multi-intelligent agents can complete the assigned tasks in a complex environment. However, the method does not take into consideration the clustering results and the carrying capacity of the robots.]”

Where in the revised manuscript this change can be found [page 2, and L56-80.]

Comments 4: [There is no "Discussion" section, it should usually be there.]

Response 4: We would like to thank the reviewer for this important comment.

We have added the “Discussion” section in the revised version.

The supplementary content is as follows:” [6. Discussion

Through simulation experiments and real robot experiments, the algorithm pro-posed in this paper shows good performance, firstly, according to the carrying capacity of the robot, the limitation of the maximum number of clusters is added to K-means++, so that the clustering results match with the carrying capacity of the robot, which indicates that the robot only needs to complete all the assigned tasks at one time, in addition, K-means++ compares with K-means algorithm that K-means++ algorithm is more optimized in the method of initializing the clustering centroids, which leads to better clustering results. Therefore, the k-means++ algorithm has a strong advantage over the k-means algorithm, which does not limit the maximum number of clusters, to reduce the cost of robot movement. Secondly assigning the clustering results to multiple ro-bots using PSO algorithm reduces the complexity of task assignment while increasing the accuracy of task assignment compared to task assignment using non-clustered particle swarm algorithm. In the figure it can be seen that with the non-clustered particle swarm algorithm, the task assignment is less effective resulting in a larger path cost for the robots. Finally, the particle swarm algorithm sorts each clustered task set so that each robot performs the task at the minimum cost of movement, which further im-proves the efficiency of multi-robot cooperative work.

Although the algorithm proposed in this paper has certain advantages in mul-ti-robot task allocation, there are some limitations to the method: (1) This paper does not consider the condition of task priority, which may pose challenges for the algorithm in practical applications if task priorities change; (2) This paper does not account for robot faults, and the algorithm will be unable to address the task allocation problem when robots are faulty.

In the future, we hope to incorporate task priority and dynamic change factors into the algorithm design, so that the algorithm can adjust the task allocation scheme according to actual situations, thereby improving the adaptability and robustness of the algorithm. In addition, we also aim to extend this task allocation algorithm to other fields, such as healthcare and agriculture, where its effectiveness and reliability can be validated through actual data.]”

Where in the revised manuscript this change can be found [page 19, and L502-530.]

Reviewer 2 Report

Comments and Suggestions for Authors

The authors address integration of K-means++ and PSO for modular robot task assignment. This submission presents results of simulation and work within a physical environment.

- Rewrite your study objective more precisely (line 60-63).

- Strengthen your literature review more critically. Consider adding a table that summarizes recent trends in collective robot task allocation problems. There are many studies that employs GA, PSO, and K-means++. It is still unclear of why K-means++ and PSO are combined, although you considered K-means for global task allocation and PSO for local optimization. Cite previous studies regarding hybridization of PSO and K-means.

- Provide more information about MT-SR-TA problem (ln. 230). Define the workshop facilities more specifically. Add a legend or display what is what in Figure 4. It would be good to provide a new figure that depicts the simulated tasks. Why do you choose to simulate only three mobile bots?

- line 257: Isn't it 20 tasks?

- Figure. 12: To demonstrate your approach, you should be able to compare them with more and other established heuristic algorithms. 

- Since K-means and PSO are very well known, and they (including their hybridization) have been employed a lot across many different areas, I think there might be similar study in robotics/engineering. You must claim your own novelty in the computational approach.

Author Response

Comments 1: [Rewrite your study objective more precisely (line 60-63).]

Response 1: We would like to thank the reviewer for this important comment.

We have added study objectives to the final section of the introduction to the revised manuscript

The supplementary content is as follows” [The rest of the paper is organized as follows: Section 2 describes the mathematical model for multi-robot task allocation and the framework design of the multi-robot system. Section 3 highlights the detailed process of the multi-robot task allocation algorithm. Sections 4 and 5 present the simulation experimental study and the real robot experimental study, respectively, and compare and analyze the algorithm proposed in this paper with other algorithms. Sections 6 and 7 consist of the discussion and conclusion sections of the article.]”

Where in the revised manuscript this change can be found [page 3, and L98-104.]

Comments 2: [Strengthen your literature review more critically. Consider adding a table that summarizes recent trends in collective robot task allocation problems. There are many studies that employs GA, PSO, and K-means++. It is still unclear of why K-means++ and PSO are combined, although you considered K-means for global task allocation and PSO for local optimization. Cite previous studies regarding hybridization of PSO and K-means.]

Response 2: We would like to thank the reviewer for this important comment.

We have revised the literature review and added a table summarizing trends in the collective task assignment problem for robots. The discussion section in Section 6 explains why it is important to combine K-means++ and PSO. Previous research on mixing PSO and K-means is also cited.

The supplementary content is as follows:” [Therefore, the fusion of multiple heuristic algorithms provides a new way of thinking for solving the task allocation problem. Song [20] proposed a multi-robot task allocation method based on near-field subset partitioning to solve the problem of inefficiency in the distribution of medical supplies. The algorithm first utilizes the ant colony algorithm [21] to order the task set to form a chain of tasks related to the near field. Then, an objective optimization function is designed based on the task completion time and the path cost of the robots, and a genetic algorithm is used to divide the subsets of this task chain. Then, the divided subset of tasks is assigned to each robot. Finally, the effectiveness of the algorithm is verified by simulating an application scenario in a hospital ward.

Janati [22] first used k-means to group the tasks and then allocate them. Subsequently, they used the Genetic Algorithm (GA) to optimize the clustering results. Through simulation experiments, it can be observed that this method can effectively handle a large number of tasks and address the task allocation problem for robots. However, this method does not consider the clustering results in conjunction with the carrying capacity of the robot. If the number of tasks in the clusters does not align with the processing capacity of the robot, it will result in the robot needing to incur a larger movement cost, thereby reducing the robot's efficiency in completing the tasks. Sumana [23] proposed a task assignment method based on nearest neighbor search and integrated it with path planning for multi-intelligent agents to effectively address the task assignment problem in dynamic environments. The task is assigned to neighboring multi-intelligent agents after clustering using the k-means algorithm and integrating it with the path planning of the Particle Swarm Optimization (PSO) algorithm. Simulation experiments demonstrate that, using this method, the multi-intelligent agents can complete the assigned tasks in a complex environment. However, the method does not take into consideration the clustering results and the carrying capacity of the robots. Table 1 summarizes the future directions and trends in the development of commonly used algorithms for multi-robot task allocation.

Table 1. Future directions and trends in the multi-robot task allocation problem

Task allocation optimization algorithm

Future research directions and trends

Market-based task allocation

1. The construction of a robust and reliable communication network is a necessary condition for market tasking, which has not yet been addressed.

Task allocation based on heuristic algorithms

1. There is a wide variety of heuristic algorithms, each with its own advantages and disadvantages. Future research could explore how to integrate different heuristic algorithms to fully utilize their respective advantages and improve the efficiency and quality of task allocation.

Clustering-based task allocation

1. How to determine the optimal number of tasks in a cluster is a direction for further research.

2. Developing effective switching strategies between clusters to cope with uncertainties such as robot failures is a direction for further research.

Alternative methods of allocating tasks

1. Fully apply artificial intelligence, reinforcement learning and other technologies to optimize task allocation to improve the autonomous decision-making ability and adaptability of the robot.

2. The use of multiple algorithm fusion strategy for task allocation is also a future research direction

23.  Biswas, S.; Anavatti, S. G.; Garratt, M. A. Nearest Neighbour Based Task Allocation with Multi-Agent Path Planning in Dynamic Environments. In 2017 International Conference on Advanced Mechatronics, Intelligent Manufacture, and Industrial Automation (ICAMIMIA); IEEE, 2017; pp 181–186. https://doi.org/10.1109/ICAMIMIA.2017.8387582.

]”

Where in the revised manuscript this change can be found [page 2-3, 19,21 and L55-84,502-530,610-612]

Comments 3: [Provide more information about MT-SR-TA problem (ln. 230). Define the workshop facilities more specifically. Add a legend or display what is what in Figure 4. It would be good to provide a new figure that depicts the simulated tasks. Why do you choose to simulate only three mobile bots?]

Response 3: We would like to thank the reviewer for this important comment.

We have provided a detailed description of “MT-SR-TA” in the revised manuscript, and have modified Figure 4 by adding information describing the tasks in the new figure and explaining why three robots are used in this paper.

The supplementary content is as follows:” [Gerkey and Mataric [29] categorize the multi-robot task assignment problem based on three dimensions: robot type, task type, and assignment type, specifically into the following categories:

(1) Robot type: A single-tasking robot (ST) is one that can only perform one task at a time; a multi-tasking robot (MT) is one that can perform multiple tasks simultaneously.

(2) Task types: Single-robot tasks (SR) are those that require only one robot to complete; multi-robot tasks (MR) are those that require multiple robots to complete.

(3) Assignment types: Instantaneous assignment (IA) refers to a situation where each robot is assigned one task without future planning; time-expanded assignment (TA) refers to a scenario where a series of tasks can be assigned to a robot within the planning horizon.

Therefore, based on the classifications along the above three dimensions, the robot task assignment problems can be categorized into eight distinct types: ST-SR-IA, ST-SR-TA, MT-SR-IA, MT-SR-TA, ST-MR-IA, ST-MR-TA, MT-MR-IA, and MT-MR-TA. Note that in the context of this paper, the multi-robot task assignment problem belongs specifically to the multi-tasking single-robot time-expanded assignment (MT-SR-TA) category, meaning that each robot is capable of performing multiple tasks simultaneously, but each task within this category is still assumed to be completed by only one robot (despite being multi-tasking, the robot does not collaborate with others on a single task).

A multi-robot system is a system composed of two or more robots. Due to the limitation of experimental equipment, there are only three experimental robots in our group at present, so we performed multi-robot task assignment based on these three mobile robots.]”

Where in the revised manuscript this change can be found [page 7-9, and L273-291、311.]

Comments 4: [line 257: Isn't it 20 tasks?]

Response 4: We would like to thank the reviewer for this important comment.

The supplementary content is as follows:” [The total number of tasks set in this paper is 20, but each time the robot goes to perform a task, it randomly selects some of the tasks from these 20 tasks. We have done 5 separate experiments in the paper and the number of tasks is different for each experiment.]”

Comments 5: [Figure. 12: To demonstrate your approach, you should be able to compare them with more and other established heuristic algorithms.]

Response 5: We would like to thank the reviewer for this important comment.

We have added a set of comparison experiments in the revised manuscript.

The supplementary content is as follows:” [We have added a set of comparison experiments to the revised manuscript and compared the relevant parameters with the algorithm proposed in this paper.

]”

Where in the revised manuscript this change can be found [page 11, and L357.]

Comments 6: [Since K-means and PSO are very well known, and they (including their hybridization) have been employed a lot across many different areas, I think there might be similar study in robotics/engineering. You must claim your own novelty in the computational approach.]

Response 6: We would like to thank the reviewer for this important comment.

We have explained the advantages of the algorithm used in this paper in the information in the “Discussion” section of the revised manuscript.

The supplementary content is as follows:”[Through simulation experiments and real robot experiments, the algorithm pro-posed in this paper shows good performance, firstly, according to the carrying capacity of the robot, the limitation of the maximum number of clusters is added to K-means++, so that the clustering results match with the carrying capacity of the robot, which indicates that the robot only needs to complete all the assigned tasks at one time, in addition, K-means++ compares with K-means algorithm that K-means++ algorithm is more optimized in the method of initializing the clustering centroids, which leads to better clustering results. Therefore, the k-means++ algorithm has a strong advantage over the k-means algorithm, which does not limit the maximum number of clusters, to reduce the cost of robot movement. Secondly assigning the clustering results to multiple robots using PSO algorithm reduces the complexity of task assignment while increasing the accuracy of task assignment compared to task assignment using non-clustered particle swarm algorithm. In the figure it can be seen that with the non-clustered particle swarm algorithm, the task assignment is less effective resulting in a larger path cost for the robots. Finally, the particle swarm algorithm sorts each clustered task set so that each robot performs the task at the minimum cost of movement, which further im-proves the efficiency of multi-robot cooperative work.]”

Where in the revised manuscript this change can be found [page 19, and L502-530.]

Reviewer 3 Report

Comments and Suggestions for Authors

The paper describes experiments with using a k-means clustering variant and PSO for optimizing multi-agent task allocation.

The paper is in general very well written and organized. The experiments and results are coherent with the methods and convincing.  

Here are some more detailed suggestions to hopefully help improve the paper:

  - There are some extra spaces as well as missing spaces. Careful proof-reading is recommended.

  - It's not common to mention "Literature [x] ...".  A more formal and common approach is to mention the authors' names rather than just "Literature".

  - Line 110:  SLAM is a method rather than an algorithm.

  - There could be references and possibly more details about the algorithms used, namely k-mean++, PSO, MAA, gmapping.

  - Line 213: "Some scholars ..." is quite vague for the sentence.  One or more references would substantiate the claim.

  - In Section 5, "Physics experiments" is arguably an adequate expression, since it conveys the meaning of Experimental physics applications rather than tests with physical robots. A better wording is recommended (Experiments with physical agents, real world robots, or just the robot names and types, for example)

  - There could be more details also on how the robot maps and trajectory lines were obtained/generated.

Comments on the Quality of English Language

Careful proof reading required.

Author Response

Comments 1: [There are some extra spaces as well as missing spaces. Careful proof-reading is recommended.]

Response 1: We would like to thank the reviewer for this important comment.

We have double-checked some of the extra spaces as well as missing spaces in our revised manuscript

The supplementary content is as follows” [Modifications such as “Yuan 1, *、System (ROS). The、L Є {1, 2, ..., nl},、d (Ri, Cj)、d (Ri, Tm nm)”]”

Where in the revised manuscript this change can be found [page 1、3、4and L4、22、126-127、141、144]

Comments 2: [It's not common to mention "Literature [x] ...". A more formal and common approach is to mention the authors' names rather than just "Literature]

Response 2: We would like to thank the reviewer for this important comment.

We have changed “Literature [x]” to “Author's name [x]” in the revised manuscript.

The supplementary content is as follows:” [Amend Literature [16]” to read “Zou [16]”、 Amend “Literature [17]” to read “Ren [17]”、 Amend “Literature [20]” to read “Janati [22]”

]”

Where in the revised manuscript this change can be found [page 1、2 and L40、44、65]

Comments 3: [Line 110:  SLAM is a method rather than an algorithm.]

Response 3: We would like to thank the reviewer for this important comment.

We have changed “SLAM algorithm” to “SLAM method” in the revised manuscript

The supplementary content is as follows:” [The SLAM (Simultaneous Localization and Mapping) method]”

Where in the revised manuscript this change can be found [page 4, and L154-155.]

Comments 4: [There could be references and possibly more details about the algorithms used, namely k-mean++, PSO, MAA, gmapping.]

Response 4: We would like to thank the reviewer for this important comment.

We have provided references behind the (k-mean++, PSO, gmapping) algorithms in the revised manuscript.

The supplementary content is as follows:” [k-mean++ [27], k-mean [26], PSO [25], Gmapping [24]]”

Where in the revised manuscript this change can be found [page 5, and L167、172、178、182.]

Comments 5: [Line 213: "Some scholars ..." is quite vague for the sentence.  One or more references would substantiate the claim.]

Response 5: We would like to thank the reviewer for this important comment.

We have changed “Some scholars ...” to “Author's name [x]” in the revised manuscript

The supplementary content is as follows:” [Luo [28] execute the tasks in a cluster according to the serial number of the task set within that cluster

References: Luo, X.D. Multi-robot Path Planning and Scheduling for Intelligent Warehouse. M.S., Chongqing University of Posts and Telecommunications,2022. https://doi.org/10.27675/d.cnki.gcydx.2022.000682.]”

Where in the revised manuscript this change can be found [page 7, and L257-258]

Comments 6: [In Section 5, "Physics experiments" is arguably an adequate expression, since it conveys the meaning of Experimental physics applications rather than tests with physical robots. A better wording is recommended (Experiments with physical agents, real world robots, or just the robot names and types, for example)]

Response 6: We would like to thank the reviewer for this important comment.

We have changed “Physics experiments” to “Real robot experiments” in the revised manuscript

The supplementary content is as follows:” [All “Physics experiments” in the revised manuscript have been changed to “Real robot experiments”.]”

Where in the revised manuscript this change can be found [page 14-16, and L410-412、421、470、488、491.]

Comments 7: [There could be more details also on how the robot maps and trajectory lines were obtained/generated.]

Response 7: We would like to thank the reviewer for this important comment.

We've already gone into more detail on how to get robot maps and trajectory lines.

The supplementary content is as follows:” [Among them, generating the robot's moving trajectory is a complex and crucial process that involves path planning and motion control. The path planning and motion control for the robot, used in this paper, are integrated within the move_base function package of ROS. Path planning involves searching for the optimal path from the starting point to the target point on the constructed map, using a path planning algorithm. This algorithm usually takes into account the length of the path, obstacles, and other relevant information to find the best solution. Motion control is the process of controlling the robot's mobile chassis to follow the planned path, resulting in the generation of a moving trajectory.

The Gmapping algorithm is based on a particle filter, a probabilistic filter utilizing Monte Carlo methods. It estimates the robot's state by maintaining a set of particles, each representing a possible position of the robot and the corresponding map feature. During the robot's movements, the Gmapping algorithm updates the weights of the particles based on the environmental information scanned by the LiDAR and the robot's movements, thereby enabling the estimation of the robot's position and the construction of the map [24].]”

Where in the revised manuscript this change can be found [page 11、15, and L338-346、423-429.]

Round 2

Reviewer 1 Report

Comments and Suggestions for Authors

Thanks to the authors for the corrections made. I have no more comments.